# ALKBH5-mediated m6A modification of circFOXP1 promotes gastric cancer progression by regulating SOX4 expression and sponging miR-338-3p
Shouhua Wang [1,2,3] ✉, Xiang Zhu[1,2,3], Yuan Hao[1,2,3], Ting ting Su[1,2] & Weibin Shi [1,2] ✉

Circular RNAs (circRNAs) have recently been suggested as potential functional modulators of cellular physiology processes in gastric cancer (GC). In this study, we demonstrated that circFOXP1 was more highly expressed in GC tissues. High circFOXP1 expression was positively associated with tumor size, lymph node metastasis, TNM stage, and poor prognosis in patients with GC. Cox multivariate analysis revealed that higher circFOXP1 expression was an independent risk factor for disease-free survival (DFS) and overall survival (OS) in GC patients. Functional studies showed that increased circFOXP1 expression promoted cell proliferation, cell invasion, and cell cycle progression in GC in vitro. In vivo, the knockdown of circFOXP1 inhibited tumor growth. Mechanistically, we observed ALKBH5-mediated m6A modification of circFOXP1 and circFOXP1 promoted GC progression by regulating SOX4 expression and sponging miR-338-3p in GC cells. Thus, our findings highlight that circFOXP1 could serve as a novel diagnostic and prognostic biomarker and potential therapeutic target for GC.

Gastric cancer (GC) ranks as the third leading cause of cancer-related death worldwide[1]. In recent decades, there have been considerable improvements in the early diagnosis and treatment of GC via radical resection; however, the 5-year survival rate of GC patients remains low[2,3]. Therefore, it is necessary to identify more effective biomarkers and therapeutic targets for GC diagnosis and treatment.

Circular RNAs (circRNAs) are a group of noncoding, covalently uninterrupted loop transcripts, most of which have yet to be functionally characterized[4]. There are various functions of circRNAs: sponging miRNAs or proteins; acting as scaffolds; serving as templates for translation; and regulating mRNA translation and stability[5]. In recent years, an increasing amount of evidence has shown that circRNAs are recognized as master regulators of various biological processes and key players by sponging miRNAs in GC[6,7]. For example, higher exosomal circSHKBP1 expression in gastric cancer could promote tumor progression by regulating the miR-582-3p/HUR/VEGF axis and suppressing HSP90 degradation[8]. Reduced circCUL2 expression in GC tissues and cells was found and circCUL2 regulated gastric cancer malignant transformation and cisplatin sensitivity by affecting autophagy activation through miR-142-3p/ROCK2[9]. Ectopic

expression of METTL14 markedly suppressed GC cell growth and invasion and mediated the m6A modification of circORC5, which suppressed tumor progression by regulating the miR-30c-2-3p/AKT1S1 axis[10]. The ectopic circular RNA circDLG1 expression promoted GC progression and anti-PD-1 resistance via the regulation of CXCL12 expression sponging to miR-141-3p[11].

In our previous study, we described and demonstrated that circFOXP1 (hsa_circ_0008234) expression was significantly increased in gallbladder cancer and enhanced tumor progression and the Warburg effect in gallbladder cancer via regulating PKLR expression[12]. However, the expression and function of circFOXP1 in GC have not been fully elucidated. The sponging function, also known as the competitive endogenous (CE) function, is the most mature function of circRNAs[13]. Herein, we focused mainly on the circRNA-microRNA code and investigated how this relationship impacts the regulation of circFOXP1 expression in GC.

Recently, an accumulating body of studies has revealed the mutual regulatory effects of m6A modifications and circRNAs and found that N6-methyladenosine (m6A)-driven endogenous ncRNA translation has a series of impacts on tumor progression[14]. In tumors, the m6A level of several

[1]Department of General Surgery, Xinhua Hospital, Shanghai Jiao Tong University School of Medicine, Shanghai 200092, China. [2]Shanghai Key Laboratory of Biliary Tract Disease Research, Shanghai 200092, China. [3]These authors contributed equally: Shouhua Wang, Xiang Zhu, Yuan Hao. ✉e-mail: wshlife1987@126.com; shiweibinxinhua@126.com

endogenous circRNAs was tested, and the results showed that the m6A motif was abundant in circRNAs and that m6A modification regulated the function of circRNAs[15]. Currently, the roles of m6A in circRNAs are mainly related to several factors, including the biogenesis of circRNAs, cytoplasmic export of circRNAs, degradation of circRNAs, and translation of circRNAs. m6A modification not only regulates the biogenesis and function of circRNAs but is also affected by circRNAs[16]. However, it remains unclear how m6A modification regulates circFOXP1 in GC; therefore, further research is necessary to elucidate the underlying mechanism.

In this study, we found that circFOXP1 expression was significantly upregulated in GC tissues and was positively associated with tumor size, lymph node metastasis, advanced TNM stage, and poor prognosis. A functional study showed that circFOXP1 knockdown in GC cells inhibited cell proliferation, invasion, and cell cycle progression. In vivo, circFOXP1 knockdown inhibited tumor growth. A mechanistic study showed that ALKBH5-mediated m6A modification of circFOXP1 promoted gastric

cancer progression by regulating SOX4 expression and sponging miR-338-3p, resulting in a promoting effect on GC progression.

## Results

### A high circFOXP1 expression level predicts poor prognosis in patients with GC

To elucidate the potential role of circFOXP1 in GC progression, firstly we analyzed circFOXP1 expression in human GC tissue samples compared with adjacent normal tissue samples. Our results verified that circFOXP1 expression was significantly upregulated in GC tissue samples compared to adjacent normal tissue samples (Fig. 1a). CircFOXP1 expression was classified as higher (circFOXP1 expression rate > mean expression rate) or lower (circFOXP1 expression rate < mean expression rate). Then, we analyzed the correlation between circFOXP1 expression and clinical features in GC patients. The results indicated that upregulation of circFOXP1 expression was positively associated with tumor size, lymph node metastasis, and

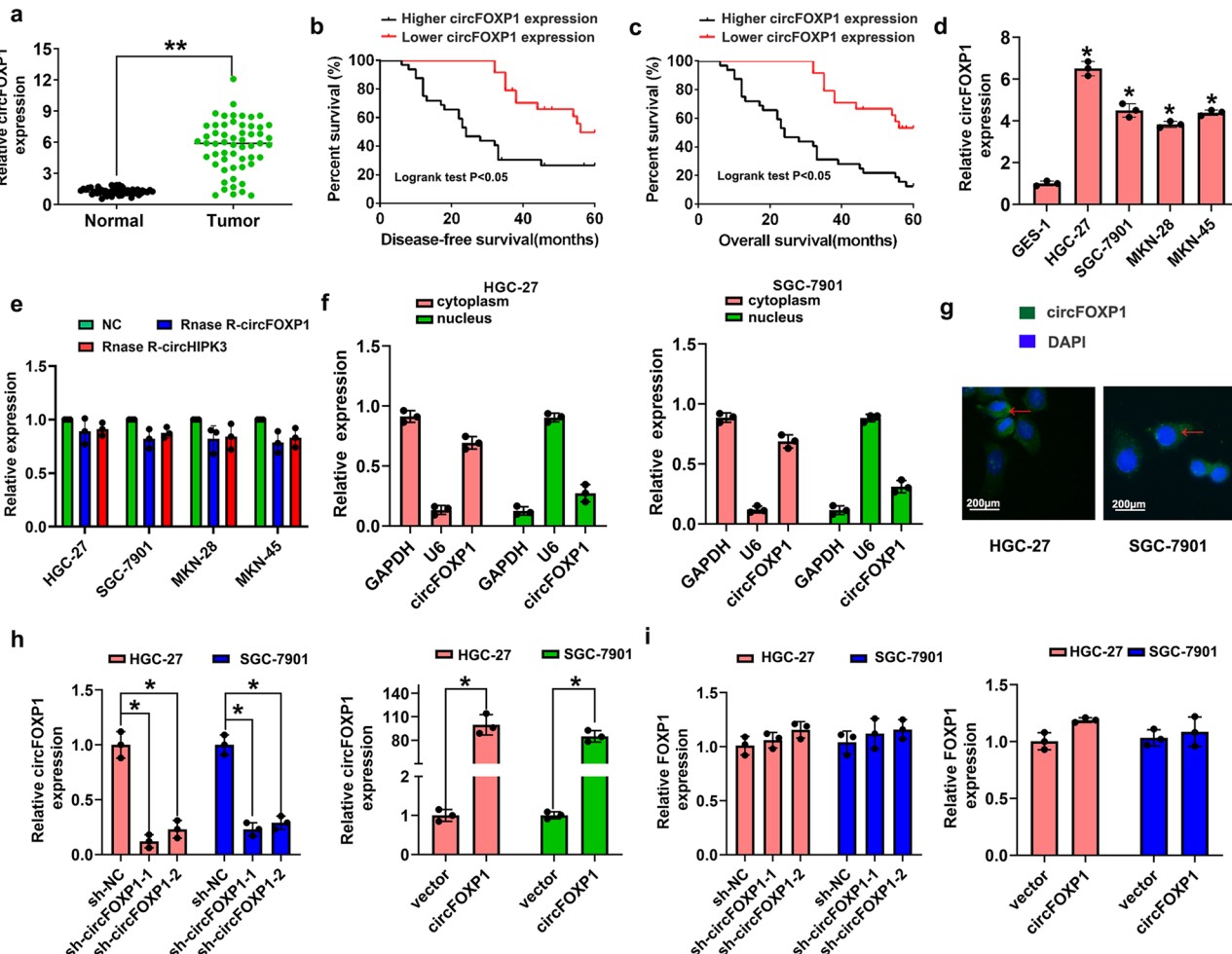

**Fig. 1 | circFOXP1 expression is upregulated in GC tissues. a** circFOXP1 expression levels were evaluated by qRT-PCR in human tissues from 56 cases of GC compared with adjacent normal tissues. The expression of circFOXP1 was normalized to GADPH. Significant differences between groups were analyzed with a paired samples *t*-test. **b, c** Kaplan-Meier analysis and log-rank tests were performed to analyze the association between the expression of circFOXP1 and DFS or OS time of GC patients. CircFOXP1 expression levels were classified as higher expression (circFOXP1 expression rate > mean expression rate) and lower expression (circFOXP1 expression rate < mean expression rate). **d** CircFOXP1 expression levels were evaluated in four human GC cell lines (HGC-27, MKN-45, MKN-28, and SGC-7901) and GES-1 cells. **e** QRT-PCR was performed to detect the circFOXP1 from control RNA, circHIPK3, or digested RNAs using RNase R exonuclease in four GC cell lines. **f** The relative mRNA expression of circFOXP1 in the nucleus or cytoplasm

in HGC-27 or SGC-7901 cells was determined by qRT-PCR. GAPDH was used as cytoplasm control and U1 was used as nuclear control. **g** RNA fluorescence in situ hybridization (FISH) was performed for circFOXP1 in HGC-27 and SGC-7901 cells. Nuclei were stained with 4, 6-diamidino-2-phenylindole (DAPI); Scale bar = 200 μm. **h** The relative mRNA expression of circFOXP1 after transfected with sh-NC, sh-circFOXP1-1 or sh-circFOXP1-2 in HGC-27 and SGC-7901 cells (left) or transfected with pLCDH-vector or pLCDH-circFOXP1 in HGC-27 and SGC-7901 cells (right). **i** The relative mRNA expression of FOXP1 after transfected with sh-NC, sh-circFOXP1-1 or sh-circFOXP1-2 in HGC-27 and SGC-7901 cells (left) or transfected with pLCDH-vector or pLCDH-circFOXP1 in HGC-27 and SGC-7901 cells (right). Data are shown as means ± SD (ANOVA or Student's *t*-test), (**d–f, h, i**) *n* = 3 for each group, *P < 0.05, **P < 0.01.

**Table 1 | The association between circFOXP1 expression level and clinicopathological characteristics in 56 cases of GC patients**

| Clinicopathological characteristics | The number of patients (n = 56) | circFOXP1 expression | | P-value |
| --- | --- | --- | --- | --- |
| | | Lower (n = 24) | Higher (n = 32) | |
| Age | | | | 0.530 |
| ≤55 | 23 | 11 | 12 | |
| >55 | 33 | 13 | 20 | |
| Gender | | | | 0.817 |
| Male | 27 | 12 | 15 | |
| Female | 29 | 12 | 17 | |
| Tumor size | | | | 0.010* |
| <3 cm | 24 | 15 | 9 | |
| ≥3 cm | 32 | 9 | 23 | |
| Histological grade | | | | 0.147 |
| Well and moderately | 36 | 18 | 18 | |
| Poorly | 20 | 6 | 14 | |
| Lymph node metastasis | | | | 0.029* |
| Negative (N0) | 17 | 11 | 6 | |
| Positive (N1-3) | 39 | 13 | 26 | |
| TNM stage | | | | 0.006* |
| I–II | 17 | 12 | 5 | |
| III–IV | 39 | 12 | 27 | |

*$P < 0.05$. TNM tumor-node-metastasis.

**Table 2 | Multivariate Cox analysis of the Disease-Free Survival (DFS) in 56 GC patients**

| Factors | Multivariate Cox analysis | | |
| --- | --- | --- | --- |
| | HR | 95% CI | P-value |
| Age | 0.978 | 0.589–1.513 | 0.943 |
| Gender | 1.123 | 0.256–1.446 | 0.486 |
| Tumor size | 1.566 | 0.885–3.231 | 0.154 |
| Histological grade | 1.006 | 0.443–1.964 | 0.501 |
| Lymph node metastasis | 2.412 | 1.166–4.188 | 0.002* |
| TNM stage | 2.209 | 1.055–3.957 | 0.003* |
| Higher circFOXP1 expression | 2.988 | 1.799–5.088 | 0.001* |

*$P < 0.05$, HR Hazard Ratio, CI Confidence intervals.

**Table 3 | Multivariate Cox analysis of the Overall Survival (OS) in 56 GC patients**

| Factors | Multivariate Cox analysis | | |
| --- | --- | --- | --- |
| | HR | 95% CI | P-value |
| Age | 0.866 | 0.422-1.609 | 0.987 |
| Gender | 1.254 | 0.355-1.567 | 0.433 |
| Tumor size | 1.433 | 0.785-2.996 | 0.122 |
| Histological grade | 1.106 | 0.677-2.066 | 0.521 |
| Lymph node metastasis | 2.212 | 1.044-4.554 | 0.003* |
| TNM stage | 2.617 | 1.322-5.257 | 0.001* |
| Higher circFOXP1 expression | 2.566 | 1.422-4.972 | 0.001* |

*$P < 0.05$, HR Hazard Ratio, CI Confidence intervals.

line (GES-1) and GC cell lines (HGC-27, SGC-7901, MKN-28, and MKN-45) by qRT-PCR analysis. The results showed that circFOXP1 was more highly expressed in four GC cell lines than in GES-1 cells (Fig. 1d). Next, we confirmed that circFOXP1 was resistant to RNase R after RNA digestion using RNase R exonuclease (Fig. 1e), which was consistent with the findings of a previous study[17]. CircFOXP1 was also found to be localized in the cytoplasm and nucleus but was predominantly enriched in the cytoplasm in HGC-27 and SGC-7901 cells (Fig. 1f). RNA-FISH assays indicated that circFOXP1 was located mainly in the cytoplasm in HGC-27 and SGC-7901 cells (Fig.1g).

Next, to confirm the functional roles of circFOXP1 in GC, we established circFOXP1-knockdown HGC-27 and SGC-7901 cells via shRNA and circFOXP1-overexpressing HGC-27 and SGC-7901 cells via overexpression plasmids (Fig. 1h). Two GC cell lines (HGC-27 and SGC7901) were selected for further investigation because they had the highest cirFOXP1 expression levels among the four GC cell lines and good transfection efficiency. Neither circFOXP1 knockdown nor overexpression affected the expression of the linear RNA FOXP1 in HGC-27 and SGC-7901 cells (Fig. 1i). To further explore the biological significance of circFOXP1 in GC progression, the proliferative ability of GC cells was assessed via CCK8 assays. As expected, circFOXP1 knockdown markedly inhibited the proliferative ability of HGC-27 and SGC-7901 cells compared to that of the corresponding control cells (Fig. 2a). However, circFOXP1 overexpression markedly promoted the proliferative ability of HGC-27 and SGC-7901 cells (Fig. 2b). Flow cytometry revealed that, compared with those in the control group, circFOXP1 knockdown caused a decrease in the S-phase and an increase in the G1 phase in HGC-27 and SGC-7901 cells (Fig. 2c). However, circFOXP1 overexpression caused an increase in the S-phase and a decrease in the G1 phase in HGC-27 and SGC-7901 cells (Fig. 2d). We also detected proliferating cell nuclear antigen (PCNA) expression, and the results showed that circFOXP1 knockdown markedly inhibited PCNA expression in HGC-27 and SGC-7901 cells compared to that in the control group. However, circFOXP1 overexpression caused increased expression of PCNA in HGC-27 and SGC-7901 cells (Fig. 2e and Supplementary Fig. 1). Transwell cell invasion assays revealed that circFOXP1 knockdown decreased the number of invasive HGC-27 and SGC-7901 cells compared to that in the control group (Fig. 2f). However, circFOXP1 overexpression increased the number of invasive HGC-27 and SGC-7901 cells (Fig. 2g).

To evaluate the biological function of circFOXP1 in vivo, a xenograft tumor model was constructed by inoculating different clones of SGC-7901 cells subcutaneously into nude mice. The results confirmed that the mean tumor volume and weight were lower and that tumor growth was reduced in the circFOXP1-knockdown group compared with those in the control group (Fig. 2h–j). These results indicated that circFOXP1 downregulation inhibited GC growth in vivo.

advanced TNM stage (Table 1, $P < 0.05$). Furthermore, Kaplan-Meier analysis revealed that patients with higher circFOXP1 expression had poorer disease-free survival (DFS) and overall survival (OS) than those with lower circFOXP1 expression (Fig. 1b, c). In addition, multivariate Cox analysis revealed that lymph node metastasis, advanced TNM stage, and increased circFOXP1 expression were risk factors for poor disease-free survival (DFS) (Table 2, $P < 0.05$) and overall survival (OS) (Table 3, $P < 0.05$) in patients with GC. Therefore, these results indicated that circFOXP1 might be a prognostic marker for GC patients.

**CircFOXP1 promotes GC cell proliferation and invasion in vitro, and the knockdown of circFOXP1 inhibits tumor growth in vivo**

Given that circFOXP1 expression is obviously increased in GC tissues, we speculated that circFOXP1 may function as a tumor gene in GC progression. We detected the mRNA levels of circFOXP1 in a normal gastric epithelial cell

**CircFOXP1 expression is negatively correlated with miR-338-3p expression and regulates miR-338-3p expression in GC**

According to our previous study and miRbase (https://www.mirbase.org/), circFOXP1 has the potential to bind with miR-338-3p. The binding sites of

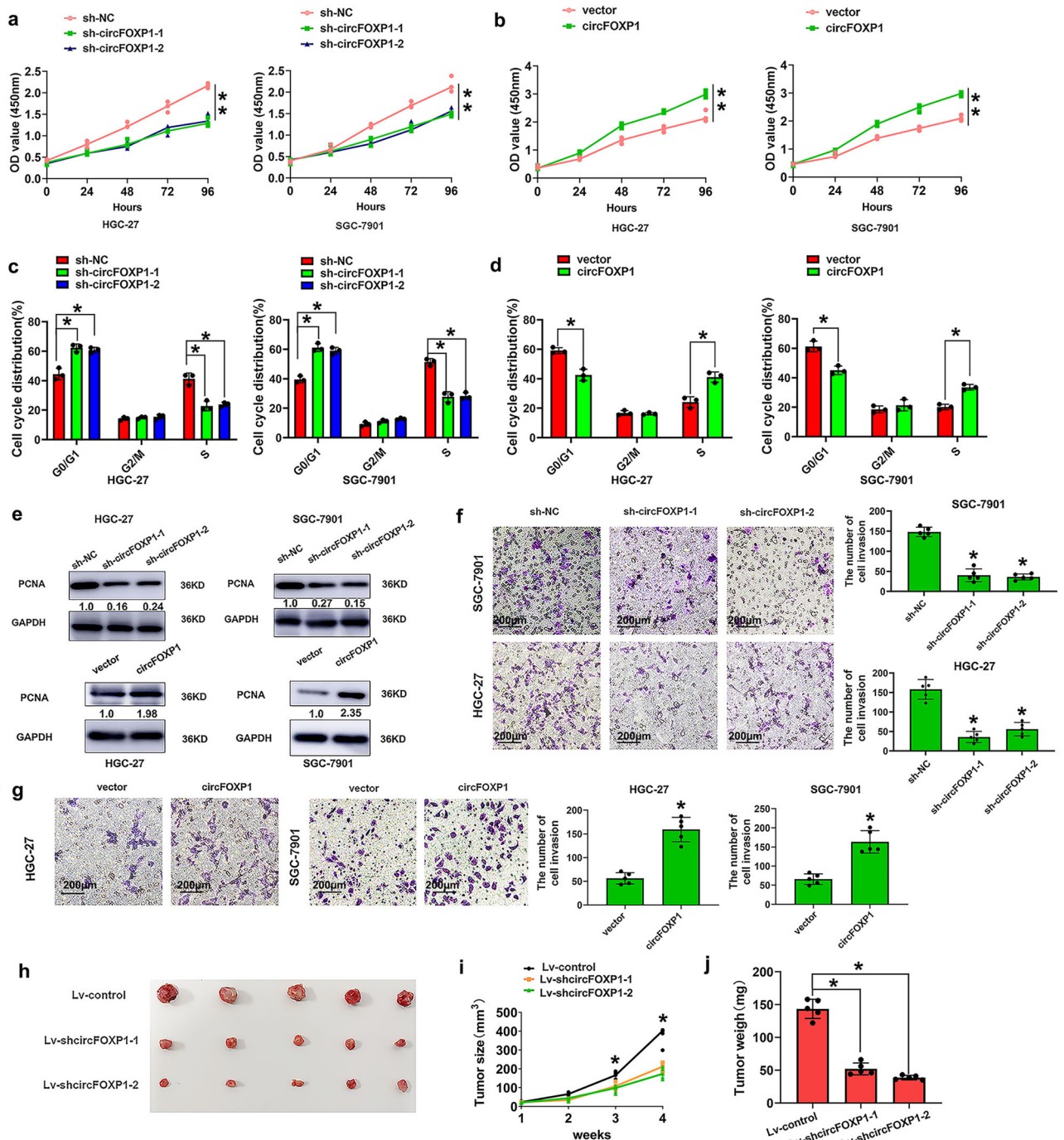

**Fig. 2 | circFOXP1 promotes GC cell proliferation, cell cycle, and cell invasion in vitro and knockdown of circFOXP1 inhibits tumor growth in vivo. a** The cell proliferation capacity was evaluated by CCK8 assays. Briefly, 2000 cells/well were plated in triplicate, and cell proliferation was detected at 0, 24, 48, 72, and 96 hours after transfection of HGC-27 or SGC-7901 cells with sh-NC, sh-circFOXP1-1 or sh-circFOXP1-2. **b** Cell proliferation capacity was evaluated with CCK8 assays. Briefly, 2000 cells/well were plated in triplicate, and cell proliferation was detected at 0, 24, 48, 72, and 96 hours after transfection of HGC-27 or SGC-7901 cells with pLCDH-vector or pLCDH-circFOXP1. **c** Data are presented as the percentage cell phase distribution including G0/G1, S, and G2/M phases after transfection of HGC-27 or SGC-7901 cells with sh-NC, sh-circFOXP1-1 or sh-circFOXP1-2 cells. **d** Data are presented as the percentage cell phase distribution including G0/G1, S, and G2/M phases after transfection of HGC-27 or SGC-7901 cells with pLCDH-vector and

pLCDH-circFOXP1. **e** circFOXP1 knockdown remarkably inhibited PCNA expression in HGC-27 or SGC-7901 cells compared to the control group (up). Besides, circFOXP1 overexpression remarkably enhanced PCNA expression in HGC-27 or SGC-7901 cells compared to the control group (down). **f** The data are presented as cell invasion ability and invasive cell number after transfection of HGC-27 or SGC-7901 cells with sh-NC, sh-circFOXP1-1 or sh-circFOXP1-2, Scale bar = 200 μm. **g** The data are presented as invasion ability and invasive cell number after transfection of HGC-27 or SGC-7901 cells with pLCDH-vector and pLCDH-circFOXP1, Scale bar = 200 μm. **h–j** Tumor volume and weight were detected to monitor tumor growth in subcutaneous implantation mouse models; mice were implanted with SGC-7901 cells transfected with lv-sh-NC, lv-sh-circFOXP1-1 or lv-sh-circFOXP1-2. Data are shown as means ± SD (ANOVA or Student's *t*-test), (**a–e**) *n* = 3 for each group, (**f–j**) *n* = 5 for each group, *$P < 0.05$, **$P < 0.01$.

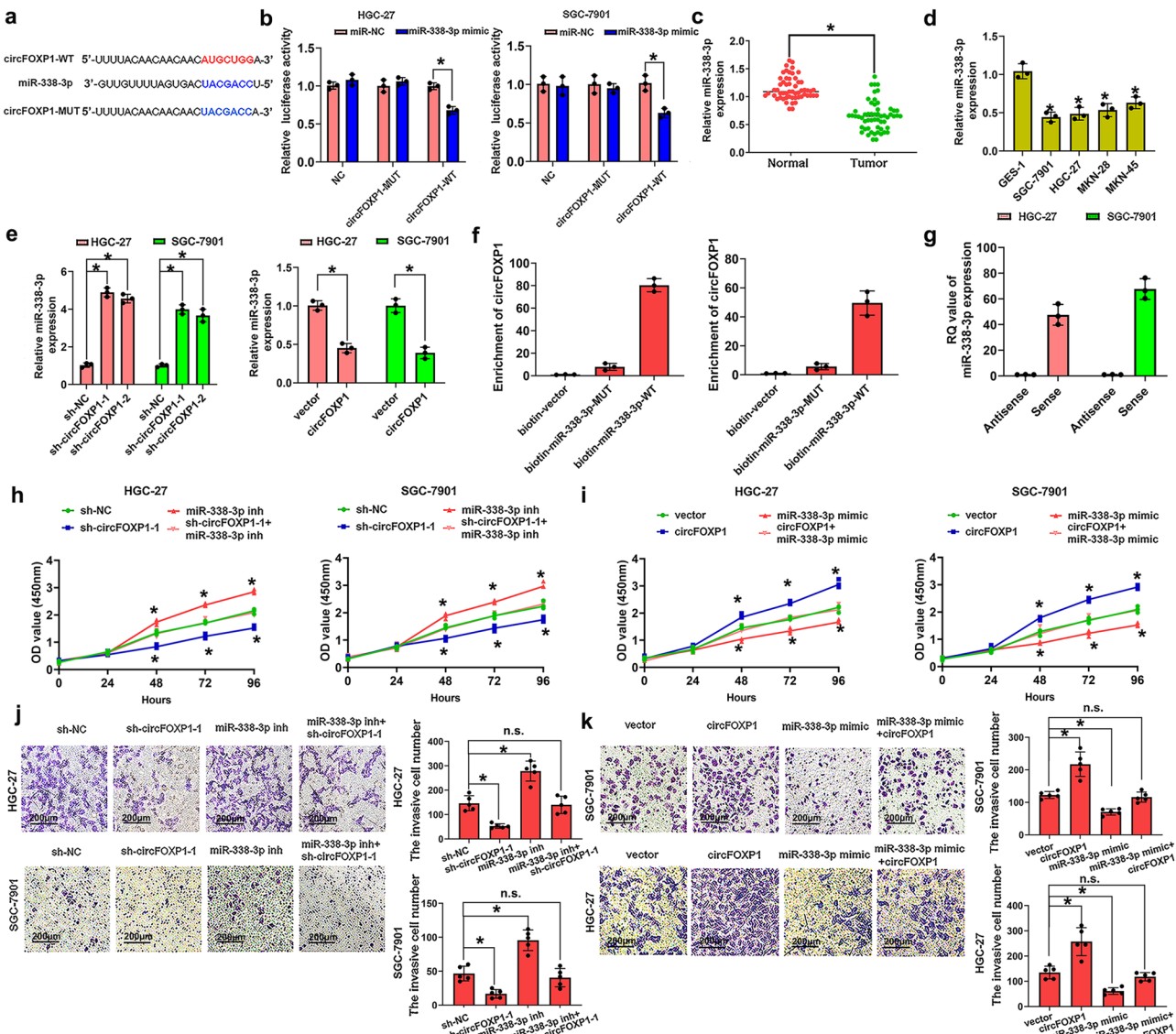

**Fig. 3 | circFOXP1 promoted GC cell proliferation and invasion by regulating miR-338-3p expression in GC. a** MiR-338-3p have complementary base pairing with circFOXP1 using circRNAs from RNA sequencing targeted miRNAs predicted by the online software tools circinteractome (http://circinteractome.nia.nih.gov). The wild-type and mutant-type complementary sequences of the circFOXP1 and miR-338-3p binding sequences are shown. **b** Luciferase reporter assays were performed in HGC-27 or SGC-7901 cells co-transfected with miR-338-3p mimic or miR-NC and circFOXP1-WT or circFOXP1-MUT reporter plasmids. **c** MiR-338-3p expression levels were evaluated by using qRT-PCR analysis in tissues from 56 cases of GC compared with adjacent normal tissues. The expression of miR-338-3p was normalized to U6. **d** MiR-338-3p expression levels were evaluated in human GC cell lines (HGC-27, MKN-45, MKN-28 and SGC-7901) and GES-1 cells by qRT-PCR analysis. The expression of miR-338-3p was normalized to U6. **e** The relative expression of miR-338-3p was detected after transfection of HGC-27 and SGC-7901 cells with sh-NC, sh-circFOXP1-1, or sh-circFOXP1-2 or was detected after transfection of HGC-27, and SGC-7901 cells with pLCDH-vector and pLCDH-circFOXP1. **f** circFOXP1 was pulled down by a miR-338-3p biotin probe in HGC-27 and SGC-7901 cells. **g** Compared with the antisense group, miR-338-3p in

circFOXP1 group (sense group) was significantly abundant. A Histogram of qRT-PCR of RNA pull-down assay was shown. **h** Cell proliferation capacity was evaluated with CCK8 assays. Briefly, 2000 cells/well were plated in triplicate, and cell proliferation was detected at 0, 24, 48, 72, and 96 hours after transfection of HGC-27 and SGC-7901 cells with sh-NC, sh-circFOXP1-1, miR-338-3p inhibitor or sh-circFOXP1-1+ miR-338-3p inhibitor. **i** Cell proliferation capacity was evaluated with CCK8 assays. Briefly, 2000 cells/well were plated in triplicate and cell proliferation was detected at 0, 24, 48, 72, and 96 hours after transfection of HGC-27 and SGC-7901 cells with pLCDH-vector, pLCDH-circFOXP1, miR-338-3p mimic or pLCDH-circFOXP1+miR-338-3p mimic. **j** The data are presented as invasion ability and invasive cell number after transfection of HGC-27 and SGC-7901 cells with sh-NC, sh-circFOXP1-1, miR-338-3p inhibitor or sh-circFOXP1-1+ miR-338-3p inhibitor, Scale bar = 200 μm. **k** The data are presented as invasion ability and invasive cell number after transfection of SGC-7901 and HGC-27 cells with pLCDH-vector, pLCDH-circFOXP1, miR-338-3p mimic or pLCDH-circFOXP1 +miR-338-3p mimic, Scale bar = 200 μm. Data are shown as means ± SD (ANOVA or Student's $t$-test), (**b**, **d**–**i**) $n = 3$ for each group, (**j**, **k**) $n = 5$ for each group, *$P < 0.05$.

miR-338-3p with circFOXP1 are shown in Fig. 3a. We found that the miR-338-3p mimic could reduce the luciferase activity of the WT circFOXP1 3'-untranslated region (UTR) but had no effect on that of the MUT circFOXP1 3'-UTR compared with that of the miR-NC group in HGC-27 and SGC-7901 cells (Fig. 3b). We analyzed the expression levels of miR-338-3p in GC tissues and confirmed that miR-338-3p expression was most significantly

decreased in GC tissues (Fig. 3c). Similarly, the expression levels of miR-338-3p in GC cells, such as HGC-27, SGC-7901, MKN-28 and MKN-45 cells, were lower than those in normal GC cells (GES-1) (Fig. 3d). qRT-PCR analysis indicated that the expression of miR-338-3p was markedly increased after circFOXP1 knockdown in HGC-27 and SGC-7901 cells compared with that in the control group (Fig. 3e, left). However, the

expression of miR-338-3p was significantly lower in HGC-27 and SGC-7901 cells after circFOXP1 overexpression than in the corresponding control cells (Fig. 3e, right). Furthermore, we performed RNA pulldown in HGC-27 and SGC-7901 cells and investigated the endogenous expression levels of circFOXP1 by using a biotin miR-338-3p probe and qRT-PCR analysis, which indicated that circFOXP1 had a stronger interaction with miR-338-3p than control cells (Fig. 3f). In addition, we confirmed that miR-338-3p was significantly more abundant in the circFOXP1 sense group than in the antisense group by RNA pull-down assay using the circFOXP1 probe in HGC-27 and SGC-7901 cells (Fig. 3g). These results demonstrated that circFOXP1 interacted with miR-338-3p in GC cells.

Furthermore, functional assays showed that circFOXP1 knockdown markedly inhibited cell proliferation compared to that in the control group, which was reversed by the miR-338-3p inhibitor in HGC-27 and SGC-7901 cells (Fig. 3h). However, compared with that in the control group, circFOXP1 overexpression markedly enhanced the proliferative ability of HGC-27 and SGC-7901 cells, which was reversed by the miR-338-3p mimic (Fig. 3i). Transwell cell invasion assays revealed that circFOXP1 knockdown decreased the number of invasive HGC-27 and SGC-7901 cells compared to that in the control group, which was reversed by the miR-338-3p inhibitor (Fig. 3j). However, compared with that in the control group, circFOXP1 overexpression markedly enhanced the invasive ability of HGC-27 and SGC-7901 cells, which was reversed by the miR-338-3p mimic (Fig. 3k). Taken together, these results indicated that circFOXP1 regulated cell proliferation and invasion via miR-338-3p in GC.

### CircFOXP1 expression regulates the miR-338-3p/SOX4 axis in GC

Increased SOX4 expression promoted GC progression, and SOX4 was identified as a target of miR-338-3p in a previous study[18]. We analyzed the expression levels of SOX4 mRNA in GC and found that SOX4 was most significantly increased in GC tissues compared to adjacent normal tissues (Fig. 4a). A schematic representation of the potential binding sites of miR-338-3p is shown in Fig. 4b. We further found that, compared with the miR-NC, circFOXP1 expression could enhance the luciferase activity of the WT SOX4 3' UTR but had no effect on that of the MUT SOX4 3'UTR. Furthermore, compared with the miR-NC, the miR-338-3p mimic reduced the luciferase activity of the WT SOX4 3' UTR but had no effect on that of the MUT SOX4 3'UTR in HGC-27 and SGC-7901 cells (Fig. 4c, d). Moreover, the qRT-PCR results indicated that circFOXP1 knockdown considerably downregulated SOX4 mRNA expression, and this effect could be reversed by the miR-338-3p inhibitor in HGC-27 and SGC-7901 cells (Figs. 4e, f). However, circFOXP1 overexpression considerably upregulated SOX4 mRNA expression, and this effect was reversed by the miR-338-3p mimic in HGC-27 and SGC-7901 cells (Fig. 4g, h). In addition, the western blot results indicated that circFOXP1 knockdown considerably downregulated SOX4 protein expression in HGC-27 and SGC-7901 cells, and these effects could be reversed by the miR-338-3p inhibitor (Fig. 4i and Supplementary Fig. 2). However, circFOXP1 overexpression considerably upregulated SOX4 protein expression in HGC-27 and SGC-7901 cells, and this effect was reversed by the miR-338-3p mimic (Fig. 4j and Supplementary Fig. 2). These findings suggested that circFOXP1 could regulate the miR-338-3p/SOX4 axis in GC.

### Knockdown of SOX4 inhibits GC cell proliferation and invasion

We also detected SOX4 protein in GC tissues and compared its expression with that in adjacent normal tissues by western blot assays. The results showed that SOX4 protein expression was significantly greater in tumor tissues than in adjacent normal tissues (Fig. 5a and Supplementary Fig. 3). Furthermore, we downregulated the expression of SOX4 in HGC-27 and SGC-7901 cells by transfection with si-SOX4 (Fig. 5b and Supplementary Fig. 3). CCK8 assay results showed that cell proliferation was inhibited by downregulating SOX4 expression in HGC-27 and SGC-7901 cells (Fig. 5c, d). Transwell cell invasion assays revealed that SOX4 knockdown decreased the number of invasive HGC-27 and SGC-7901 cells compared to that in the

control group (Fig. 5e). Flow cytometry revealed that SOX4 knockdown caused a decrease in the S-phase and an increase in the G1 phase in HGC-27 and SGC-7901 cells compared to those in the control group (Fig. 5f). These results showed that knockdown of SOX4 inhibited GC cell proliferation and invasion.

### ALKBH5 mediates m6A modification of circFOXP1 in GC

N6-Methyladenosine (m6A) is the most common posttranscriptional modification of RNA and plays a critical roles in cancer pathogenesis[19]. We analyzed the m6A modification of circFOXP1 using circPrimer and the SRAMP prediction server (http://www.cuilab.cn/sramp/), and we detected many m6A modification sites in circFOXP1. Motif analysis of the circFOXP1 methylation site based on different confidence levels of SRAMP is shown in Fig. 6a. Previous studies demonstrated that the a-ketoglutarate-dependent dioxygenase ALKB homologue 5 (ALKBH5) (a demethylase) can remove m6A methylation from its target RNAs and lead to decreased levels of m6A[20]. We hypothesized that ALKBH5 may be the major upstream dominator of circFOXP1 in GC. The results of qRT-PCR analysis revealed that ALKBH5 overexpression could increase circFOXP1 expression in HGC-27 and SGC-7901 cells (Fig. 6b, c and Supplementary Fig. 4). However, ALKBH5 knockdown decreased circFOXP1 expression in HGC-27 and SGC-7901 cells (Fig. 6d, e and Supplementary Fig. 4). Next, we performed RIP assays and confirmed that ALKBH5 could bind to circFOXP1 in HGC-27 and SGC-7901 cells (Fig. 6f). M6A RNA immunoprecipitation (MeRIP) assays were applied to explore which m6A regulators participated in the modulation of circFOXP1. Primers were designed to target this position for MeRIP detection. MeRIP assays indicated that potential m6A-modified segments of circFOXP1 could be enriched by anti-m6A rather than by anti-IgG (Fig. 6g). Moreover, ALKBH5 overexpression decreased circFOXP1 m6A levels in HGC-27 and SGC-7901 cells (Fig. 6h). To confirm that ALKBH5-mediated modulation was m6A dependent, we constructed recombinant luciferase reporter plasmids by inserting partial sequences of circFOXP1 with wild-type or mutated m6A sites. When ALKBH5 was disrupted, the luciferase activity of GC cells transfected with wild-type plasmids was attenuated, while the activity of the mutant group remained unchanged (Fig. 6i). Thus, these results indicated that ALKBH5 mediates the m6A modification of circFOXP1 in GC.

### Discussion

A large amount of circRNAs have been validated through high-throughput sequencing and bioinformatics methods in mammalian cells[21]. Recently, increasing research evidence has unveiled that many circRNAs are aberrantly expressed in various cancers, indicating their crucial roles in tumor occurrence and development. CircRNAs can serve as promising biomarkers for cancer diagnosis and prognosis[22]. In GC, circRNAs have also been found to act as tumor suppressors or oncogenes through different molecular mechanisms[23]. In this study, our results declared that circFOXP1 expression was notably upregulated in human GC tissue samples compared to adjacent normal tissue samples. We found that upregulation of circFOXP1 was positively correlated with tumor size, lymph node metastasis, and advanced TNM stage in GC patients. Kaplan-Meier analysis revealed that patients with higher circFOXP1 expression had poorer prognosis. The above results indicated that circFOXP1 could have strong potential as a diagnostic, prognostic, and predictive biomarker for GC patients. In a previous study, circFOXP1 was reported to be associated with several human diseases, including human tumors. CircFOXP1 is derived from FOXP1 and is an oncogene. For example, the expression of the circular RNA circ-FOXP1 was remarkably upregulated in hepatocellular carcinoma (HCC) tissues, and this upregulated circFOXP1 was induced by SOX9, which promoted HCC progression through sponging to miR-875-3p and miR-421[24]. Another published report suggested that downregulated circFOXP1 inhibited cell proliferation, migration, invasion, and the Warburg effect in renal cell carcinoma. Additionally, circFOXP1 was induced by ZNF263 upregulated U2AF2 expression to accelerate tumorigenesis and the Warburg effect through sponging miR-423-5p[25]. In osteosarcoma, circFOXP1 expression

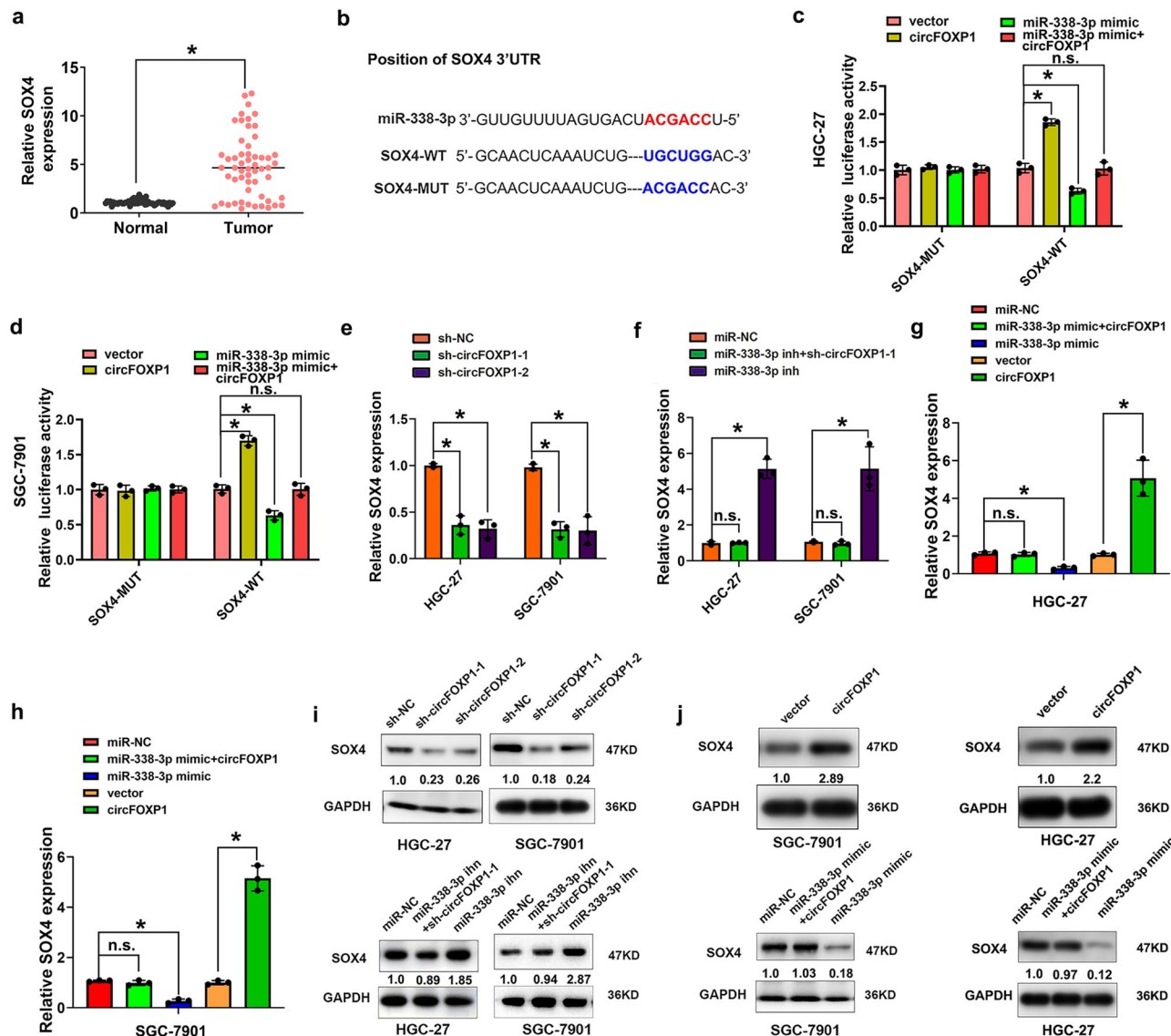

**Fig. 4 | circFOXP1 regulates miR-338-3p/SOX4 axis in GC. a** SOX4 expression levels were evaluated with qRT-PCR in tissues from 56 cases of GC compared with adjacent normal tissues. The expression of circFOXP1 was normalized to GADPH. **b** The wild-type and mutant-type complementary sequences of the SOX4 and miR-338-3p binding sequence are shown. **c, d** Luciferase reporter assays were performed in HGC-27 and SGC-7901 cells transfected with pLCDH-vector, pLCDH-circFOXP1, miR-338-3p mimic or co-transfected with miR-338-3p mimic+pLCDH-circFOXP1 and SOX4-WT or SOX4-MUT reporter vector. **e, f** The relative mRNA expression of SOX4 was detected after transfection of HGC-27 and SGC-7901 cells with sh-NC, sh-circFOXP1-1 or sh-circFOXP1-2 or miR-NC, miR-338-3p inhibitor or co-transfection with miR-338-3p inhibitor and sh-circFOXP1-1. **g, h** The relative mRNA expression of SOX4 was detected after transfection of HGC-27 and SGC-7901 cells with pLCDH-vector, pLCDH-circFOXP1 or miR-NC, miR-338-3p mimic, co-transfection with pLCDH-circFOXP1 and miR-338-3p mimic. **i** The relative protein expression of SOX4 was detected after transfection of HGC-27 and SGC-7901 cells with sh-NC, sh-circFOXP1-1 or sh-circFOXP1-2 or miR-NC, miR-338-3p inhibitor or co-transfection with miR-338-3p inhibitor and sh-circFOXP1-1. **j** The relative protein expression of SOX4 was detected after transfection of HGC-27 and SGC-7901 cells with pLCDH-vector, pLCDH-circFOXP1 or miR-NC, miR-338-3p mimic, co-transfection with pLCDH-circFOXP1 and miR-338-3p mimic. Data are shown as means ± SD (ANOVA or Student's *t*-test), (**c–j**) *n* = 3 for each group, **P* < 0.05.

was also increased and promoted angiogenesis by directly binding to microRNA-127-5p and regulating CDKN2AIP expression[26]. circFOXP1 expression was increased in lung adenocarcinoma (LUAD) and promoted cell proliferation by interacting with the miR-185-5p and regulating WNT1 expression in LUAD[27]. In colon cancer tissues, the expression of circFOXP1 and FOXP1 was negatively correlated. CircFOXP1 recruits DNMT1 to hypermethylated the promoter of FOXP1, thereby inhibiting the expression of FOXP1, and ultimately facilitating tumor progression in colon cancer[28]. In our previous study, we identified that circFOXP1 expression was remarkably upregulated in gallbladder cancer tissues and cells, and promoted tumor progression and the Warburg effect in GBC cells by regulating PKLR expression[12]. However, the clinical expression and functions of circFOXP1 in GC progression remain largely unknown.

Subsequently, functional assays demonstrated that circFOXP1 could promote GC cell proliferation, invasion, and progression. In vivo, tumor xenografts were generated in nude mice, and our results demonstrated that circFOXP1 downregulation inhibited GC growth. It is well known that exonic circRNAs are located mainly in the cytoplasm. Our FISH experiment demonstrated that circFOXP1 was present in the cytoplasm. The most commonly reported circRNAs act as tumor inhibitors or promotors through the sponges of miRNAs implicated in GC progression[4]. Subsequently, we focused on the mechanism underlying the association between circFOXP1 and GC progression in terms of expression dysregulation and regulatory mechanisms. We used these databases to predict potential miRNAs, and selected miR-338-3p was able to bind to circFOXP1. We performed dual-luciferase reporter and RNA pulldown assays in GC cells.

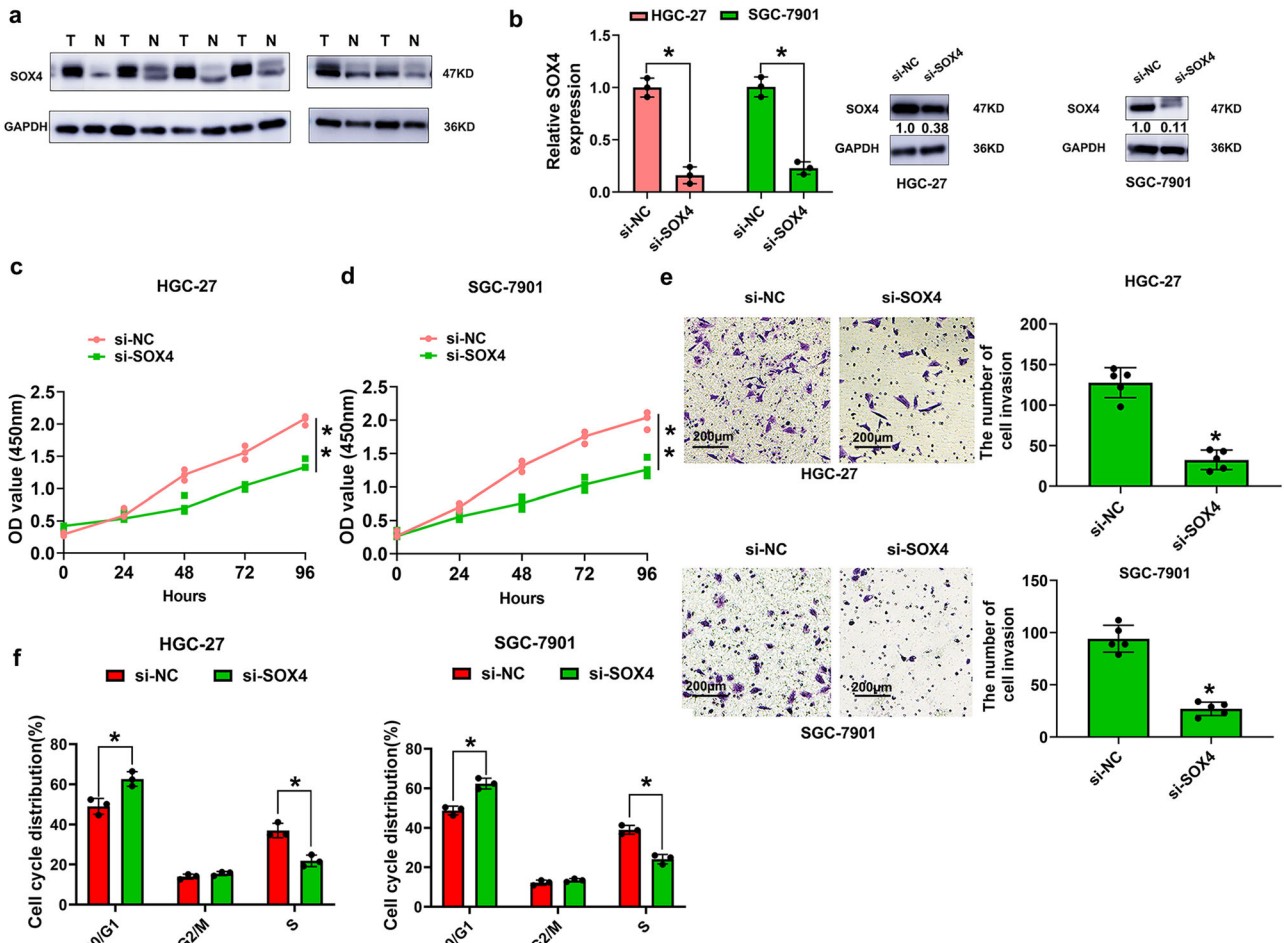

**Fig. 5 | Knockdown of SOX4 inhibits GC cell proliferation and invasion. a** SOX4 expression levels were evaluated by western blot analysis in human tissues from 56 cases of GC compared with adjacent normal tissues. **b** The relative mRNA and protein expression of SOX4 was detected after transfection of HGC-27 and SGC-7901 cells with si-NC and si-SOX4. **c, d** Cell proliferation capacity was evaluated with CCK8 assays. Briefly, 2000 cells/well were plated in triplicate, and cell proliferation was detected at 0, 24, 48, 72, and 96 hours after transfection of HGC-27 and SGC-7901 cells with si-NC or si-SOX4. **e** The data are presented as invasion ability and invasive cell number after transfection of HGC-27 and SGC-7901 cells with si-NC or si-SOX4, Scale bar = 200 μm. **f** Data are presented as the percentage cell phase distribution including G0/G1, S, and G2/M phases after transfection of HGC-27 and SGC-7901 cells with si-NC or si-SOX4. Data are shown as means ± SD (ANOVA or Student's $t$-test), (**b–d, f**) $n = 3$ for each group, **e** $n = 5$ for each group, $*P < 0.05$. $**P < 0.01$.

The results demonstrated that the endogenous circFOXP1 was pulled down from the miR-338-3p biotin probe through qRT-PCR analysis or that miR-338-3p was significantly enriched by RNA pull-down assay using the cir-cFOXP1 probe, indicating that circFOXP1 interacted with miR-338-3p in GC. To further investigate whether the biological function of circFOXP1 in GC cells was related to miR-338-3p, we conducted a rescue experiment. Our findings indicated that circFOXP1 promoted the proliferation and invasion of GC cells by regulating miR-338-3p. SOX4 was identified as a target of miR-338-3p in GC, and a schematic representation of the potential binding sites of miR-338-3p was generated[18]. Thus, we performed dual-luciferase reporter and western blotting assays and demonstrated that circFOXP1 affects SOX4 expression by regulating miR-338-3p in GC.

N6-methyladenosine is the most common and reversible internal modification of mRNAs or circRNAs, and recent studies have shown that N6-methyladenosine (m6A)-mediated cap-independent translation initiation is a potential mechanism for circRNA translation[29]. ALKBH5 is a member of the AlkB family and can remove m6A methylation from target RNAs, leading to a decrease in m6A levels[30]. An increasing number of investigations have shown that ALKBH5 acts as an important role in GC progression[31]. For example, a decreased expression of ALKBH5 was detected in GC samples, and ALKBH5 expression was correlated with clinical tumor distal metastasis and lymph node metastasis, while the demethylase ALKBH5 suppressed cell invasion of gastric cancer via the PKMYT1 m6A modification[20]. LncNRON expression was upregulated in GC tissues and exerts its oncogenic functions by binding to the N6-methyladenosine eraser ALKBH5 and mediating the decay of Nanog mRNA[32]. We investigated that whether m6A, the most abundant type of internal mRNA modification, was related to the modulation of circFOXP1. In the present study, by conducting MeRIP assays, we found that over-expressed ALKBH5 reduced total m6A and circFOXP1 m6A levels but increased circFOXP1 expression in GC cells. Subsequently, we confirmed that ALKBH5 could bind to circFOXP1 in GC cells. These results indicated that ALKBH5 mediates the m6A modification of circFOXP1 during GC progression (Fig. 7).

In conclusion, our study provided convincing in vitro and in vivo evidence that circFOXP1 functions to promote GC proliferation by regulating SOX4 expression through the sponging of miR-338-3p. In addition to demonstrating the relationship between ALKBH5 and circFOXP1, we further explored ALKBH5 mediates the m6A modification of circFOXP1 in GC, leading to GC development. We also investigated the biological mechanisms underlying the development of GC to provide assistance for future treatment opportunities.

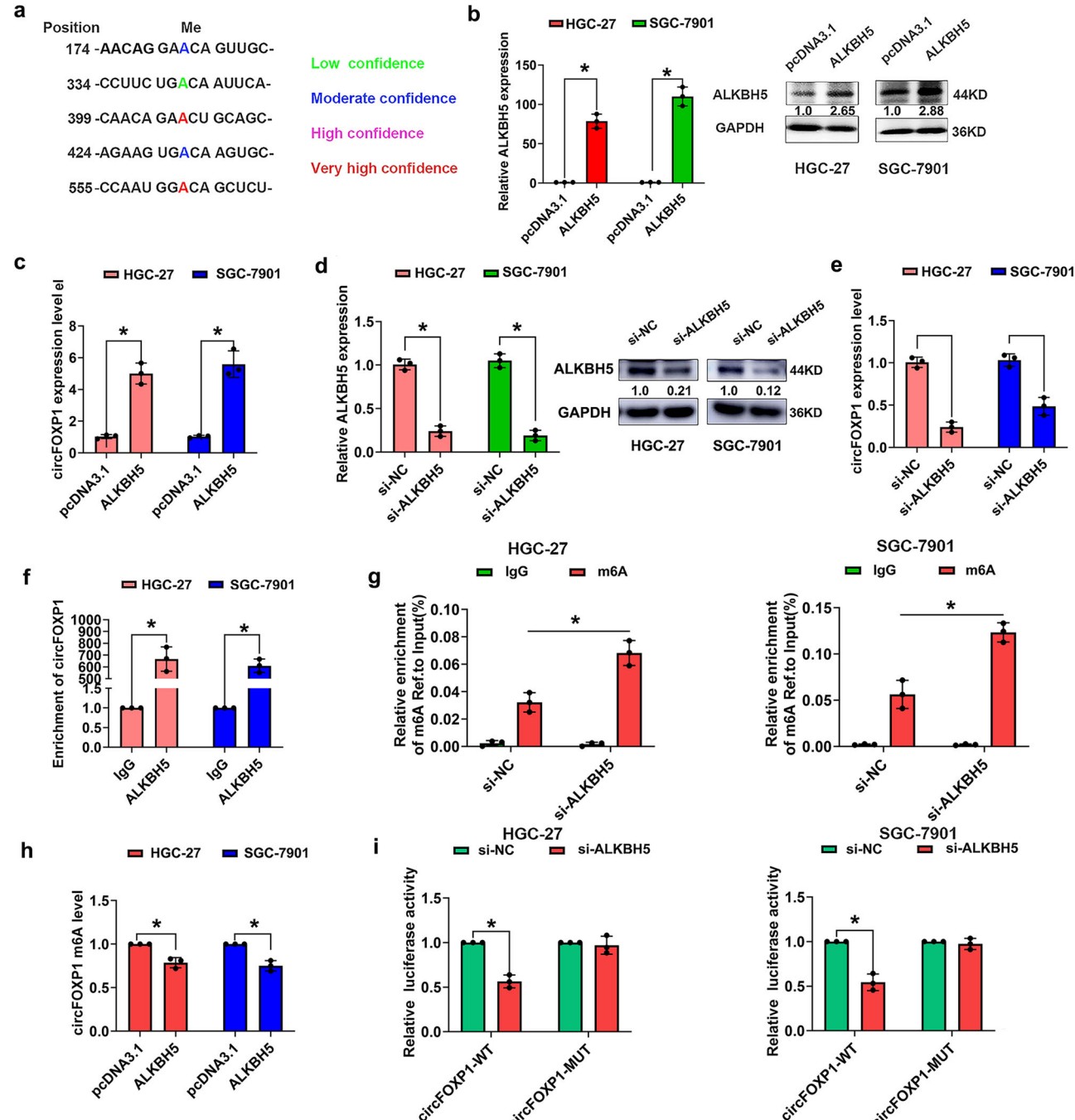

**Fig. 6 | ALKBH5 mediates m6A modification of circFOXP1 in GC. a** Prediction results of circFOXP1 in SRAMP website show the potential site of m6A modification. The red arrow points to sites with very high confidence. **b** The relative mRNA or protein expression of ALKBH5 after transfected with pcDNA3.1 or pcDNA3.1-ALKBH5 in HGC-27 and SGC-7901 cells by qRT-PCR or western blot assay. **c** The relative expression of circFOXP1 after transfected with pcDNA3.1 or pcDNA3.1-ALKBH5 in HGC-27 and SGC-7901 cells by qRT-PCR. **d** The relative mRNA or protein expression of ALKBH5 after transfected with si-NC or si-ALKBH5 in HGC-27 and SGC-7901 cells by qRT-PCR or western blot assay. **e** The relative expression of circFOXP1 after transfected with si-NC or si-ALKBH5 in HGC-27 and SGC-7901 cells by qRT-PCR. **f** RIP showed that circFOXP1 interacted with ALKBH5 in HGC-27 and SGC-7901 cells. **g** MeRIP analyses of the effects of ALKBH5 knockdown on the m6A levels in HGC-27 and SGC-7901 cells. **h** ALKBH5 overexpression reduced circFOXP1 m6A levels in HGC-27 and SGC-7901 cells. **i** Luciferase reporter assays were performed by transfected with circFOXP1-WT or circFOXP1-MUT m6A position and si-NC or si-ALKBH5 in HGC-27 and SGC-7901 cells. Data are shown as means ± SD (ANOVA or Student's *t*-test), (**b–i**) $n = 3$ for each group, *$P < 0.05$.

## Methods
### Patient tissue samples
A total of 56 snap-frozen GC tissues and paired adjacent normal tissues were collected from patients diagnosed with GC at Xinhua Hospital from May 2015 to December 2018. All the enrolled patients in this study had never received preoperative therapy, and tissue samples were collected and frozen in liquid nitrogen immediately after surgical resection. Clinicopathological characteristics, including age, sex, tumor size, histological grade, lymph node metastasis, tumor-node-metastasis (TNM) stage, survival, and recurrence, were also collected. All participants signed informed consent before this study. The study was performed in accordance with the Declaration of Helsinki and the guidelines of the Committee of the Human

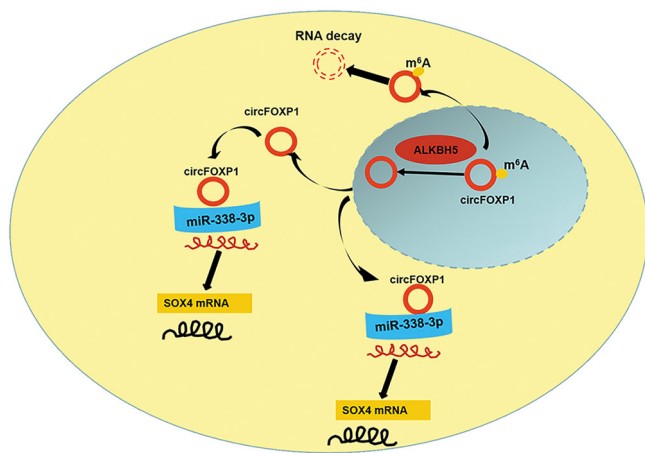

**Fig. 7 | Schematic illustration of ALKBH5/circFOXP1/miR-338-3p/SOX4 axis.** ALKBH5-mediated m6A modification of circFOXP1 expression promotes gastric cancer progression by regulating SOX4 expression and sponging miR-338-3p.

Ethics Committee of Xinhua Hospital. The Committee of the Human Ethics Committee of Xinhua Hospital approved the study protocol. All ethical regulations relevant to human research participants were followed.

## Cell line culture
Human gastric cancer cell lines (HGC-27, MKN-45, MKN-28, and SGC-7901) were purchased from the Cell Bank of the Chinese Academy of Sciences (Shanghai, China). A human normal gastric epithelial cell line (GES-1) was purchased from the Cell Bank of the Chinese Academy of Sciences (Shanghai, China). GES-1 cells were cultured in high-glucose DMEM supplemented with 10% foetal bovine serum (FBS; Invitrogen, Shanghai, China), 100 μg/mL penicillin, and 100 U/mL streptomycin. HGC-27, MKN-45, MKN-28 and SGC-7901 cells were cultured in RPMI 1640 medium (Gibco, Invitrogen, USA) supplemented with 10% FBS, 100 μg/mL penicillin and 100 U/mL streptomycin. All the cells were cultured at 37 °C in a humidified 5% $CO_2$ atmosphere.

## Cell transfection
Cell transfection was performed with Lipofectamine 3000 Reagent (Invitrogen, CA, USA) according to the manufacturer's protocol. The shRNA sequences against circFOXP1 were cloned and inserted into a pGPU6/GFP/Neo vector (GenePharma Co., Ltd., Shanghai, China). pLCDH-circFOXP1 was synthesized using full-length circFOXP1 (hsa_circ_0008234) and subcloned and inserted into a pLCDH vector with the cloning sites BamHI/EcoRI (GENESEED, Guangzhou, China). WT circFOXP1 with potential miR-338-3p binding sites and circFOXP1 with a mutation in these sites or a WT/MUT SOX4-3'UTR with potential miR-338-3p binding sites were constructed by chemosynthesis (Sangon Biotech, China) and fused to the luciferase reporter vector psiCHECK2 (Promega, Madison, WI, USA). Specific targeting of SOX4 and ALKBH5 siRNAs or overexpressing ALKBH5 plasmids was designed and synthesized by GenePharma (Shanghai, China). The sequences for the miR-338-3p mimic, miR-338-3p inhibitor, and miR-NC were also obtained from Gene Pharma (Shanghai, China). The sequences are shown in Supplementary Table 1. GC cells were plated in 6-well plates ($5 \times 10^4$/per well) 12 h prior to transfection and were transfected at 50-70% confluence. The miRNA mimic or inhibitor (100 nM/per well) was prepared, incubated at room temperature for 20 min, and mixed with Lipofectamine 3000 (Invitrogen, Carlsbad, CA, USA) according to the manufacturer's instructions.

## RNA fluorescence in situ hybridization (FISH)
Fluorescent probes for circFOXP1 labelled with Cy3 were synthesized by GenePharma (Shanghai, China). HGC-27 and SGC-7901 cells were cultured to the exponential phase and were approximately 90% confluent at the

time of fixation. Then, the cells were fixed using 4% formaldehyde for 25 min. The cells were permeabilized with 1% Triton X-100 in PBS for 30 min and then washed for 5 min. The cells were hybridized in a hybridization buffer and incubated with 4 ng/μl RNA probes at 37 °C overnight. A Leica SP5 confocal microscope (Leica Microsystems, Wetzlar, Germany) was used to detect the signals and acquire images.

## Cell Counting Kit-8 (CCK8) assay
The cells were transfected with Lipofectamine 3000 (Life Technologies) in a serum-free medium according to the manufacturer's instructions. Transfected HGC-27 and SGC-7901 cells were seeded in 96-well plates ($2 \times 10^3$ cells per well). At 0, 24, 48, 72, and 96 hours after cell transfection, 10 μl of CCK-8 solution (Dojindo Laboratories, Kumamoto, Japan) was added to each well of the plate. Then, the cells were incubated for 2 hours in the incubator. Finally, the absorbance was detected at 450 nm using a microplate reader (BioTek Instruments, Inc., Winooski, VT, USA).

## Flow cytometry analysis
After 48 h of incubation, the transfected cells were harvested, washed, and then fixed with 70% ethanol at −20 °C overnight. After RNase digestion, the cells were stained with 20 μg/ml propidium iodide (PI; Beyotime, Shanghai, China) at 37 °C for 30 min, and 100 μg/ml RNase A was subsequently added to the cells and incubated at 4 °C in the dark for 30 min. The cell cycle was examined by flow cytometry using a FACSCalibur system (BD Biosciences, San Jose, CA, USA). The data are presented as the percentage of cells in each phase, including the G0/G1, S, and G2/M phases. Gating strategy for flow cytometry in the analysis of cell cycle distribution in Supplementary Fig. 5.

## Cell invasion assays
Cell invasion assays were performed with Transwell plates (BD Falcon, USA) precoated with Matrigel in 24-well Transwell chambers with 8-mm pore polycarbonate filters (Millipore, Billerica, MA, USA). A total of $1 \times 10^5$ cells were cultured in the upper chamber in medium without serum, while 10% foetal bovine serum (FBS) was added to the lower chamber (Gibco, Invitrogen, USA). After transfection at 48 hours, the cells on the lower layer of the membrane were fixed with methanol and stained with 1% crystal violet for 30 min at room temperature. The cells were counted by using an Olympus microscope, and five fields were randomly selected for cell counting (magnification, 200×).

## Quantitative real-time PCR (QRT-PCR) analysis
Total RNA was extracted from tissues or cells using TRIzol reagent (Thermo Fisher Scientific, Rockford, IL, USA). cDNA was synthesized using a Prime Script RT reagent kit (Takara, Shiga, Japan). Gene quantification and amplification were performed by using Absolute qPCR Premix (Thermo Fisher Scientific, USA) with the StepOnePlus™ system. The mRNA expression was analyzed by using SYBR Green Real-Time PCR Master Mixes (Thermo Fisher Scientific, USA) with an ABI 7900 Fast Thermal Cycler (Applied Biosystems; Thermo Fisher Scientific, USA). GAPDH or U6 was used as a reference. The primer sequences are shown in Supplementary Table 1. The relative mRNA expression was calculated using the $2^{-\Delta\Delta Ct}$ method.

## Western blotting assays
Total protein was extracted using RIPA buffer (Beyotime, Beijing, China). Equal amounts of total protein were separated via SDS-polyacrylamide gel electrophoresis (SDS-PAGE) and then transferred onto PVDF membranes (Millipore, Billerica. MA. USA). The membrane was blocked with 5% nonfat milk and incubated with primary antibodies against SOX4 (1:500; Cell Signaling Technology, Houston, TX, USA) and ALKBH5 (1:1000; Cell Signaling Technology, Houston, TX, USA). and GAPDH (1:1000, Santa Cruz, Dallas, TX, USA) overnight at 4 °C. Next, secondary antibodies were added for 1.5 hours, after which each protein band was detected via an enhanced chemiluminescence (ECL) detection system (Amersham Biosciences, Buckinghamshire, UK).

## Luciferase reporter assays

The wild-type (WT) circFOXP1 or SOX4 3'-untranslated region (UTR) containing the miR-338-3p targeting sequence and the mutated type (MUT) were amplified and cloned and inserted into the luciferase reporter plasmid psicheck-2 vector (Promega, Madison, WI). A total of $5\times10^3$ GC cells were seeded in a 96-well plate and cotransfected with 150 ng of empty psicheck2 vector, psicheck2-circFOXP1-WT or psicheck2-circFOXP1-MUT (Sangon Biotech, Shanghai, China) or 2 ng of pRL-TK (Promega, Madison, WI, USA) in combination with a miR-338-3p mimic or miR-NC using Lipofectamine 2000 (Invitrogen, Carlsbad, California, USA). To demonstrate that SOX4 was a target of circFOXP1, $5\times10^3$ GC cells seeded in a 96-well plate were cotransfected with 150 ng of vector, pLCDH-cirFOXP1, miR-338-3p mimic or pLCDH + miR-338-3p mimic; psicheck2-SOX4-3'UTR-WT/MUT (Sangon Biotech, Shanghai, China); or 2 ng of pRL-TK (Promega, Madison, WI, USA) using Lipofectamine 2000 (Invitrogen, Carlsbad, California, USA). After cell transfection for 48 h, the luciferase activity in the cell lysates was analyzed using a Dual-Luciferase Reporter Assay System (Promega) according to the manufacturer's instructions.

For the wild-type plasmids, partial sequences of circFOXP1 containing potential m6A sites (supported by SRAMP prediction and MeRIP-qPCR) were inserted into luciferase reporter vectors. For the mutant plasmids, adenosines (A) at the m6A positions were replaced by cytosines (C). The cells were collected and lysed for luciferase detection 48 hours after transfection. Luciferase activities were detected via using the Dual-Luciferase Reporter Assay System (Promega, USA). The relative luciferase activity was normalized against the Renilla luciferase activity.

## Biotin-coupled RNA pull-down

The 3' end biotinylated miR-338-3p or control RNA was designed and synthesized by GenePharma (Suzhou, China). Cells were transfected with 50 nM biotin-labelled miRNAs (Gene Create, Wuhan, China). Streptavidin-coupled Dynabeads (Invitrogen) were washed and resuspended in a buffer, after which the biotin-labeled miRNAs were added. After incubating at room temperature for 10 min, the coated beads were separated with a magnet for 2 min. The pulled-down RNA was extracted with TRIzol reagent, followed by qRT-PCR analysis.

## In vivo xenograft experiments

Xenograft experiments ($n = 5$/per group) were conducted using 3-week-old BALB/c nude mice. All animal protocols have been approved by the Institutional Animal Care and Use Committee of Xinhua Hospital and we have complied with all relevant ethical regulations for animal use. A total of $1 \times 10^5$ SGC-7901 cells were transfected and subcutaneously injected into the flank. The tumor volume and weight were evaluated weekly as follows: tumor volume ($mm^3$) = (length) × (width)$^2$/2. After 4 weeks, the mice were sacrificed, and the tumor tissues were processed for further analysis. According to the AVMA Guidelines for the Euthanasia of Animals, all the mice were euthanized by an intraperitoneal injection of a threefold dose of barbiturates. After that, we removed the tumors immediately and measured their length, width, and weight. No mice died accidentally during feeding.

## Methylated RNA immunoprecipitation (MeRIP) and RNA binding protein immunoprecipitation assays

Total RNA extraction was conducted by using the RNAiso Plus (Takara, Shiga, Japan), after which DNase was added to remove DNA. An m6A RNA enrichment kit (EpigenTek) was used to detect MeRIP levels according to the manufacturer's instructions. The m6A-containing target fragment was pulled down using a bead-bound m6A capture Ab, after which the RNA sequence containing both ends of the m6A region was cleaved using a lyase cocktail. The enriched RNA was released, purified, and eluted. Quantitative real-time PCR (QRT-PCR) was performed after MeRIP to quantify changes in target gene m6A methylation.

Antibodies against ALKBH5 (Abcam, #ab195377) were used in RNA-binding protein immunoprecipitation (RIP) assays. RIP assays were performed using an EZ-Magna RIP™ RNA-Binding Protein Immunoprecipitation Kit (Millipore, Billerica, MA, USA) according to the manufacturer's instructions. Cells at approximately 90% confluence were lysed using complete RIP lysis buffer containing RNase Inhibitor (Millipore) and protease inhibitor, and 100 μl of whole-cell extract was subsequently incubated with RIP buffer containing magnetic beads conjugated to specific antibodies. The detailed method of the RIP assay was described in a previously published study[12].

## Statistics and reproducibility

All the statistical analyses were conducted using GraphPad Prism (GraphPad Software, Inc., La Jolla, USA). Statistical analysis was carried out using a $t$-test or Bonferroni multiple comparisons test. A chi-square test was used to analyze the differences in the distributions of the variables, and the Pearson correlation coefficient was calculated to assess correlations of expression. Survival curves of GC patients were generated using Kaplan-Meier analysis and the log-rank test. The data are expressed as the mean values ± the standard deviation (means ± SD). A P value less than 0.05 and 0.01 was considered to indicate statistical significance.

## Reporting summary

Further information on research design is available in the Nature Portfolio Reporting Summary linked to this article.

## Data availability

All other data are available from the corresponding author (or other sources, as applicable) on reasonable request. The numerical source data behind the graphs in the manuscript can be found in Supplementary Data 1. The uncropped gels are shown in Supplementary Fig. 1-4.

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

## Acknowledgements
This study was funded by the Shanghai Jiao Tong University Medical and Industrial Cross Research Fund Youth Project.

## Author contributions
W.S., S.W. and H.Y. designed the study. W.S. and Z.X. performed the majority of the experiments and wrote the original manuscript. S.T. and W.S. analyzed the data. All the authors contributed to the manuscript preparation and gave final approval of the submitted manuscript.

## Competing interests
The authors declare no competing interests.

## Ethics
All aspects of this study were approved by the Ethics Committee of Xinhua Hospital, Shanghai Jiao Tong University School of Medicine, China.
