## [Peer Review File · Communications Biology]

Reviewers' comments:

Reviewer #1 (Remarks to the Author):

The introduction is too sparsely described, the issues raised in the paper should be expanded, more literature should be cited.

The assumptions raised in the paper are novel and may be of interest to researchers working on this topic, although I'm not sure that the methods chosen are the most accurate for determining the stated goals.

Too limited information has been provided regarding basic in vitro culture. Were no antibiotics or antimicrobials used in the culture? What culture dishes were used, at what concentrations were the cells seeded into the dishes, and how long did the culture last?

The results obtained are interesting and significantly expand the knowledge of GC in humans. In addition, the detailed information presented in the results chapter may become a valuable contribution to the development of gene therapies based on non-coding RNA. However, my biggest objection is the lack of detailed description of the research methodology, which makes it impossible to reproduce the analyses performed by other researchers. Therefore, I recommend a detailed and major editing of the manuscript to be considered for publication.

Reviewer #2 (Remarks to the Author):

The study developed by Wang and colleagues aimed to demonstrate the effect of circFOXP1 overexpression in GC, correlating with clinical characteristics (clinical value). Furthermore, the authors were able to trace the relationship of circHOXP1/miR-3338-3p/SOX4 in GC; and, identified the relationship of ALKBH5 and m6A-modification in circFOXP1. All of these results were obtained through experiments considered strong evidence.

The study is interesting and well designed, and may be of interest to several colleagues working studying circRNA biology and GC. However, some results based on cell lines may bring some weaknesses to the article, for example, the results on the function of circFOXP1 on the HGC-27, MKN-28 and MKN-45 cell lines.

Below are some questions and suggestions:

1 - In the results, the authors state that "CircFOXP1 expression was also found to be localized in the cytoplasm and nucleus, but was predominately enriched in the cytoplasm in HGC-27 and SGC-7901 cells (Figure 1F)", Lines 280-282. I believe it is the opposite because according to figure 1F CircFOXP1 was predominantly enriched in the nucleus and not in the cytoplasm as the authors claim.

2 - I observed (Lines 273-311) that the authors for several times demonstrate the results of cellular assays highlighting only the SCG-7901 lineage or the HCG-27 lineage, and at other times demonstrating the results in both. I believe that the authors were unable to identify the expected result in both strains in some experiments, however, they do not make this clear in the text or through graphics. I suggest that

authors should report negative results in strains that did not demonstrate expected results in text or graphics.

3 - Another fact that I observed is the lack of results for the MKN-28 and MKN-45 lines other than those described in Lines 274-276. What happened to the results of these strains and why were only the SGC-7901 and HGC-27 strains chosen to continue the study? This has to be clear in the text.

4 - The authors state that in lines 330-334 that “Furthermore, we performed RNA pulldown in SGC-7901 cells and investigated the endogenous expression levels of circFOXP1 by using biotin miR-338-3p probe by qRT-PCR analysis, which indicated that circFOXP1 was interacted with miR-338-3p compared with those in the input control (Figure 3F-3G). These results demonstrated that circFOXP1 interacted with miR-338-3p in GC.” According to the graph, the authors used miR-338-3p-MUT (Mutated) and circFOXP1-WT (Wild Type), given the mutation in the miRNA used, would it be possible for this interaction between the wild-type circRNA and the mutated miRNA to exist?

5 - Lines 338-340: “CircFOXP1 overexpression remarkably enhanced cell proliferative capability in SGC-7901 cells compared to the control group, which was reversed by miR-338-3p-mimic (Figure 3I).” And in the HGC-27 lineage, this effect was not observed? Make it clear in the text.

6 - Lines 343-345: “However, circFOXP1 overexpression remarkably enhanced cell invasive capability in SGC-7901 cells compared to the control group, which was reversed by miR-338-3p mimic (Figure 3K).” Again, and in the HGC-27 lineage this effect was not observed? Make it clear in the text.

7 - Lines 362-364: “However, circFOXP1 overexpression considerably upregulated SOX4 mRNA expression in SGC-7901 cells, and this effect could be reversed by miR-338-3p mimic (Figure 4G).” Again, and in the HGC-27 lineage this effect was not observed? Make it clear in the text.

8 - Lines 367-369: “However, circFOXP1 overexpression considerably upregulated SOX4 protein in SGC-7901 cells, and this effect could be reversed by miR-338-3p mimic (Figure 4I).” Again, and in the HGC-27 lineage this effect was not observed? Make it clear in the text.

9 - I suggest that the results of “Knockdown of SOX4 inhibits GC cell proliferation and invasion” (Line 402) come before to “ALKBH5 mediates m6A modification of circFOXP1 in GC”, Line 372.

Dear editor:

We had revised our manuscript according to reviewers' suggestions by point to point below:

Referee expertise:

Reviewers' comments:

Reviewer #1 (Remarks to the Author):

The introduction is too sparsely described, the issues raised in the paper should be expanded, more literature should be cited.

Thanks for your suggestion, we had revised the introduction by increasing the content and citation of literature in the introduction section according to your suggestion.

The assumptions raised in the paper are novel and may be of interest to researchers working on this topic, although I'm not sure that the methods chosen are the most accurate for determining the stated goals. Too limited information has been provided regarding basic in vitro culture. Were no antibiotics or antimicrobials used in the culture? What culture dishes were used, at what concentrations were the cells seeded into the dishes, and how long did the culture last?

Thanks for your suggestion, we had checked and demonstrated the methodology section. We added accurate information regarding the use of antibiotics and cell concentrations, which seeded into the dishes, and how long in vitro culture according to your suggestions and had detailed addition of other methods section.

The results obtained are interesting and significantly expand the knowledge of GC in humans. In addition, the detailed information presented in the results chapter may become a valuable contribution to the development of gene

therapies based on non-coding RNA. However, my biggest objection is the lack of detailed description of the research methodology, which makes it impossible to reproduce the analyses performed by other researchers. Therefore, I recommend a detailed and major editing of the manuscript to be considered for publication.

Thanks for your suggestion, for different experimental sections, we have added more detailed description and methods for facilitating researchers to repeat these experiments in the Materials and Methods sections, and added the sequences in Supplementary Table 1 according to your suggestions. In addition, we revised the language by NPG Language Editing and major editing of the manuscript.

Reviewer #2 (Remarks to the Author):

The study developed by Wang and colleagues aimed to demonstrate the effect of circFOXP1 overexpression in GC, correlating with clinical characteristics (clinical value). Furthermore, the authors were able to trace the relationship of circHOXP1/miR-3338-3p/SOX4 in GC; and, identified the relationship of ALKBH5 and m6A-modification in circFOXP1. All of these results were obtained through experiments considered strong evidence.

The study is interesting and well designed, and may be of interest to several colleagues working studying circRNA biology and GC. However, some results based on cell lines may bring some weaknesses to the article, for example, the results on the function of circFOXP1 on the HGC-27, MKN-28 and MKN-45 cell lines.

Below are some questions and suggestions:

1 - In the results, the authors state that “CircFOXP1 expression was also found to be localized in the cytoplasm and nucleus, but was predominately enriched in the cytoplasm in HGC-27 and SGC-7901 cells (Figure 1F)”, Lines 280-282. I believe it is the opposite because according to figure 1F CircFOXP1 was

predominantly enriched in the nucleus and not in the cytoplasm as the authors claim.

Thanks for your suggestion, circFOXP1 expression was really found to be localized in the cytoplasm in Figure 1F, Fluorescent probes for circFOXP1 labelled with Cy3 is the green part and we revised and marked with arrows and added the detection for HGC-27 cell in Figure 1G and showed that circFOXP1 expression was really found to be localized in the cytoplasm. The nucleus is the blue part by staining with 4, 6-diamidino-2-phenylindole (DAPI). circFOXP1 expression in the nucleus is really lower than that in the cytoplasmic expression and there is very little green appearing in Figure 1G in HGC-27 and SGC-7901 cells.

2 - I observed (Lines 273-311) that the authors for several times demonstrate the results of cellular assays highlighting only the SCG-7901 lineage or the HCG-27 lineage, and at other times demonstrating the results in both. I believe that the authors were unable to identify the expected result in both strains in some experiments, however, they do not make this clear in the text or through graphics. I suggest that authors should report negative results in strains that did not demonstrate expected results in text or graphics.

Thank you for your suggestion. We added experimental results by using both HGC-27 and SGC-7901 cells in Figure 1G, Figure 1I, Figure 2B, Figure 2D, Figure 2G, Figure 3I, Figure 3K, Figure 4G, Figure 4J, and so on, and obtained expected experimental results in two GC cells according to your suggestions.

3 - Another fact that I observed is the lack of results for the MKN-28 and MKN-45 lines other than those described in Lines 274-276. What happened to the results of these strains and why were only the SCG-7901 and HCG-27 strains chosen to continue the study? This has to be clear in the text.

Thank you for your suggestion. because there are many cell lines in gastric cancer, in addition to the four cell lines we have chosen, such as MGC803, KATO-III, BGC-803, HSC-38, AGS, and so on. we only selected the expression of four cell lines, due to limited cell lines in our experimental research group. Two GC cell lines (HGC-27 and SGC7901) that often used by some other researchers, were selected for further investigation because they had the highest cirFOXP1 expression levels among the four GC cell lines and good transfection efficiency. If we replicate all four types of cells in all experiments, the workload will be enormous, and we hope to further confirm this in future work.

4 - The authors state that in lines 330-334 that “Furthermore, we performed RNA pulldown in SGC-7901 cells and investigated the endogenous expression levels of circFOXP1 by using biotin miR-338-3p probe by qRT-PCR analysis, which indicated that circFOXP1 was interacted with miR-338-3p compared with those in the input control (Figure 3F-3G). These results demonstrated that circFOXP1 interacted with miR-338-3p in GC.” According to the graph, the authors used miR-338-3p-MUT (Mutated) and circFOXP1-WT (Wild Type), given the mutation in the miRNA used, would it be possible for this interaction between the wild-type circRNA and the mutated miRNA to exist?

Thank you for your suggestion, we performed RNA pulldown in HGC-27 and SGC-7901 cells and investigated the endogenous expression levels of circFOXP1 by using a biotin miR-338-3p probe and qRT-PCR analysis, which indicated that circFOXP1 had a stronger interaction with miR-338-3p than control cells. In the experiments, indeed, there is very little circFOXP1 expression enrichment was detected between the wild-type circRNA and the mutated miRNA by qRT-PCR analysis, but it is very few expression than that between wild-type circRNA and the wild miRNA by qRT-PCR analysis. Thus, this interaction that between the wild-type circRNA and the wild miRNA exist, but this interaction between the wild-type

circRNA and the mutated miRNA do not exist. Moreover, we also added the experiments by RNA pull-down assay using the circFOXP1 probe in HGC-27 and SGC-7901 cells and confirmed that miR-338-3p was significantly more abundant in the circFOXP1 sense group than in the antisense group in Figure 3G. Thus, we further demonstrated that circFOXP1 was interacted with miR-338-3p.

5 - Lines 338-340: “CircFOXP1 overexpression remarkably enhanced cell proliferative capability in SGC-7901 cells compared to the control group, which was reversed by miR-338-3p-mimic (Figure 3I).” And in the HGC-27 lineage, this effect was not observed? Make it clear in the text.

Thank you for your suggestion, we added the experiment in HGC-27 cells, and demonstrated that circFOXP1 overexpression remarkably enhanced cell proliferative capability compared to the control group, which was reversed by miR-338-3p-mimic in Figure 3I.

6 - Lines 343-345: “However, circFOXP1 overexpression remarkably enhanced cell invasive capability in SGC-7901 cells compared to the control group, which was reversed by miR-338-3p mimic (Figure 3K).” Again, and in the HGC-27 lineage this effect was not observed? Make it clear in the text.

Thank you for your suggestion, we added the experiment in HGC-27 cells, and demonstrated that circFOXP1 overexpression remarkably enhanced cell invasion capability compared to the control group, which was reversed by miR-338-3p-mimic in Figure 3K.

7 - Lines 362-364: “However, circFOXP1 overexpression considerably upregulated SOX4 mRNA expression in SGC-7901 cells, and this effect could be reversed by miR-338-3p mimic (Figure 4G).” Again, and in the HGC-27 lineage

this effect was not observed? Make it clear in the text.

Thank you for your suggestion, we added the experiment in HGC-27 cells, and demonstrated that circFOXP1 overexpression considerably upregulated SOX4 mRNA expression, and this effect could be reversed by miR-338-3p mimic in Figure 4G.

8 - Lines 367-369: “However, circFOXP1 overexpression considerably upregulated SOX4 protein in SGC-7901 cells, and this effect could be reversed by miR-338-3p mimic (Figure 4I).” Again, and in the HGC-27 lineage this effect was not observed? Make it clear in the text.

Thank you for your suggestion, we added the experiment in HGC-27 cells, and demonstrated that circFOXP1 overexpression considerably upregulated SOX4 protein expression, and this effect could be reversed by miR-338-3p mimic in 4J.

9 - I suggest that the results of “Knockdown of SOX4 inhibits GC cell proliferation and invasion” (Line 402) come before to “ALKBH5 mediates m6A modification of circFOXP1 in GC”, Line 372.

Thank you for your suggestion, we had adjusted the order of results that knockdown of SOX4 inhibits GC cell proliferation and invasion” come before to “ALKBH5 mediates m6A modification of circFOXP1 in GC” according to your suggestion, in addition, we had also adjusted the figure order.

REVIEWERS' COMMENTS:

Reviewer #1 (Remarks to the Author):

The presented manuscript explores the topic of circular RNAs, which have been found to exhibit a number of regulatory functions. In this study, authors focused on circFOXP1 and its potential correlation with gastric cancer (GC). Their findings revealed that circFOXP1 expression is increased in GC, and this increase has implications for patient survival outcomes. The authors also associated circFOXP1 overexpression with specific clinical features such as tumor size, lymph node metastases, and TNM stages.

I would like the authors to include in the discussion the issue of the role of circFOXP1 in colorectal cancer (for example: PMID: 32951006).

I have no further comments and I recommend that the above manuscript be published.

Reviewer #2 (Remarks to the Author):

I thank the authors for their effort in responding and adjusting possible weaknesses that could bring some bias to the study. I am satisfied with the review and my opinion is that the paper is suitable for publication.

Dear editor:

We had revised our manuscript according to reviewers' suggestions by point to point below:

Reviewer #1 (Remarks to the Author):

The presented manuscript explores the topic of circular RNAs, which have been found to exhibit a number of regulatory functions. In this study, authors focused on circFOXP1 and its potential correlation with gastric cancer (GC). Their findings revealed that circFOXP1 expression is increased in GC, and this increase has implications for patient survival outcomes. The authors also associated circFOXP1 overexpression with specific clinical features such as tumor size, lymph node metastases, and TNM stages.

I would like the authors to include in the discussion the issue of the role of circFOXP1 in colorectal cancer (for example: PMID: 32951006).

I have no further comments and I recommend that the above manuscript be published.

Thanks you, we had added the discussion the issue of the role of circFOXP1 in colorectal cancer (for example: PMID: 32951006).

Reviewer #2 (Remarks to the Author):

I thank the authors for their effort in responding and adjusting possible weaknesses that could bring some bias to the study. I am satisfied with the review and my opinion is that the paper is suitable for publication.

Thanks your review for our manuscript.